# Text2CAD-Bench: A Benchmark for LLM-based Text-to-Parametric CAD Generation

## Abstract

Text-to-CAD generation aims to create parametric CAD models from natural language, enabling rapid prototyping and intuitive design workflows. However, existing benchmarks focus on basic primitives and simple sketch-extrude sequences, lacking advanced features essential for real-world applications and covering only traditional mechanical parts. We introduce Text2CAD-Bench, the first benchmark systematically evaluating text-to-CAD across geometric complexity and application diversity. Our benchmark comprises 600 human-curated examples spanning four levels: L1-L2 cover fundamental geometry with standard features, L3 introduces complex topology and freeform surfaces, and L4 extends to real-world domains beyond mechanical parts. Each example pairs dual-style prompts—geometric descriptions mimicking non-expert users, and procedural sequences aligned with expert-level conventions. Evaluating mainstream general LLMs and domain-specific models, we find that current models perform reasonably on basic geometry but degrade substantially on complex topology and advanced features. We release our benchmark to drive progress in text-to-CAD research.

## 1. Introduction

The emergence of Large Language Models (LLMs) has catalyzed significant progress in Computer-Aided Design (CAD), particularly in the automatic generation of parametric models from natural language. Unlike mesh or point cloud representations, parametric CAD maintains editable construction histories and precise dimensional constraints, making it essential for engineering design and manufacturing. Early approaches, such as DeepCAD(Wu et al.),

pioneered the field by employing sequence-based representations of sketch-and-extrude operations(Koch et al.). Recently, however, research has shifted toward code-based generation—using frameworks such as CadQuery or Python scripts (Xie & Ju). By capitalizing on the code synthesis capabilities of pre-trained LLMs(Li et al., 2023), code-based methods offer a higher level of abstraction, enabling more natural descriptions of complex geometries and parametric relationships compared to low-level command sequences.

Despite these advances, the systematic evaluation of text-to-CAD systems remains limited. Existing benchmarks exhibit three key shortcomings that lead to a misalignment between human design intent and machine-executable representation.

First, geometric complexity is heavily restricted: current datasets predominantly consist of simple sketch-extrude primitives and lack support for advanced modeling features such as chamfers, fillets, sweeps, lofts, and freeform surfaces. As a result, generated models tend to be geometrically simplistic and often inadequate for real-world manufacturing.

Second, domain coverage is narrow, with prior benchmarks focusing almost exclusively on traditional mechanical parts and basic primitives, thereby overlooking diverse applications in areas such as medical devices, consumer products, and architectural fabrication.

Third, linguistic diversity is insufficient and structurally biased. Descriptions in existing benchmarks frequently mirror low-level command sequences rather than convey high-level design intent. This encourages models to learn a superficial "command-translation" mapping instead of achieving genuine "shape-understanding," ultimately failing to capture how humans naturally describe shapes—whether through visual-geometric attributes or procedural construction logic.

To bridge these gaps, we introduce Text2CAD-Bench, the first comprehensive benchmark for evaluating text-to-CAD generation across a rigorously defined complexity hierarchy. Our benchmark consists of 600 human-curated examples, organized into four progressive tiers from L1 to L4. L1–L3 systematically evaluate geometric reasoning from basic primitives to advanced topological operations, such as sweeping a profile along a curved path. L4 extends the

---

[1]Anonymous Institution, Anonymous City, Anonymous Region, Anonymous Country. Correspondence to: Anonymous Author <anon.email@domain.com>.

Preliminary work. Under review by the International Conference on Machine Learning (ICML). Do not distribute.

*Figure 1.* Overview of the Text2CAD-Bench construction pipeline. Our human-in-the-loop process comprises four stages: (1) Design Specification defines target geometry and complexity levels; (2) Collaborative Authoring combines human expertise with AI assistance to develop validated CadQuery code; (3) Description Generation creates dual-style prompts (geometric and sequence) for each model; (4) Quality Assurance ensures executability, description-geometry correspondence, and cross-style consistency.

evaluation to diverse real-world domains, requiring models to interpret geometric specifications within concrete application contexts—for instance, designing a medical brace or a consumer electronics accessory. This structure not only tests a model's ability to handle increasingly complex geometry, but also examines its generalization beyond conventional mechanical parts.

Our contributions are summarized as follows:

- We construct **Text2CAD-Bench**, a dataset of 600 high-quality parametric CAD models organized into four difficulty levels, specifically addressing the lack of advanced geometric operations (chamfers, fillets, sweeps, lofts, complex surfaces) and application diversity in prior benchmarks.

- We design dual-style prompts—geometric descriptions mimicking non-expert users and command sequences aligned with expert-level CAD conventions—and empirically demonstrate that geometric-style prompts yield superior generation performance across evaluated models.

- Through comprehensive evaluation of mainstream LLMs and domain-specific models, we reveal that current state-of-the-art systems struggle significantly on advanced features (L3) and real-world scenarios (L4), establishing Text2CAD-Bench as a challenging target for future research.

## 2. Related Work

### 2.1. CAD Generation Methods

Deep learning approaches to CAD generation have evolved significantly in recent years. DeepCAD pioneered the representation of CAD models as sequences of sketch-and-extrude operations, training a Transformer-based autoencoder on 178K models(Khan et al.). This sequence-based paradigm was subsequently extended by SkexGen (Xu et al., b), which disentangled topology and geometry into separate codebooks to address posterior collapse, HNC (Xu et al., a) introduced hierarchical code trees enabling controllable generation and ExtrudeNet(Ren et al., 2022) focus on the research of extrude operation.

An alternative line of work operates directly on boundary representations (B-rep)(Xu et al., 2024). SolidGen (Jayaraman et al.) proposed an autoregressive model that synthesizes B-rep geometry without requiring construction sequence supervision. Despite their progress(Seff et al., 2022), these methods share a common limitation: they require training specialized networks from scratch on task-specific representations, unable to leverage the vast knowledge encoded in pretrained language models.

### 2.2. Text-to-CAD Generation

Text-to-3D generation has witnessed rapid progress in recent years. Early work such as Text2Shape (Jun & Nichol, 2023) pioneered the mapping from natural language to 3D shapes using joint embeddings. More recently, DreamFusion (Poole et al., 2022) leveraged score distillation sampling to optimize Neural Radiance Fields (NeRF) from text prompts, enabling high-fidelity 3D content creation. Point-E (Nichol et al.) and Shap-E (Jun & Nichol, 2023) from Ope-

nAI further accelerated generation by directly producing point clouds and implicit representations, respectively.

However, these methods generate mesh or implicit representations that lack the parametric structure and dimensional constraints required for engineering applications. Text-driven CAD generation has emerged as a promising direction, with approaches categorized by their output representation(Poole et al., 2022).

Sequence-based methods generate task-specific command sequences. Text2CAD (Khan et al.) introduced the first large-scale text-CAD dataset by augmenting DeepCAD models with template-generated textual descriptions across four prompt levels. A Transformer decoder maps text embeddings to CAD command sequences, which are then converted to 3D models via a pretrained decoder.

Code-based methods leverage pretrained LLMs to generate executable CAD scripts. LLM4CAD (Li et al.) explored multimodal inputs for CadQuery code generation, finding that text-only input outperforms image-based or multimodal combinations. Text-to-CadQuery (Xie & Ju) demonstrated that fine-tuning LLMs on 170K text-CadQuery pairs yields consistent improvements with model scale. CAD-Coder (Hui et al.) further incorporated chain-of-thought reasoning and geometric reward-based reinforcement learning to enhance code validity. In the reverse engineering domain, CAD-Recode (Rukhovich et al.) showed that even small LLMs, when combined with a lightweight point cloud projector, can generate high-quality CadQuery code for CAD reconstruction.

We adopt CadQuery as our target representation for several reasons: (1) it directly leverages LLMs' code generation capabilities; (2) its method-chaining API naturally aligns with natural language descriptions; (3) it supports advanced features (chamfers, fillets, sweeps, lofts) beyond sketch-extrude; and (4) generated code is immediately executable for validation.

### 2.3. CAD Datasets and Benchmarks

Several datasets have been proposed for CAD-related research. ABC (Koch et al.) provides one million CAD models with parametric surface representations but lacks construction sequences. DeepCAD (Wu et al.) and Fusion 360 Gallery (Willis et al.) include construction sequence information but are limited to sketch-extrude operations. Text2CAD (Khan et al.) is the only existing dataset with text annotations, but its descriptions are template-generated from underlying command sequences, lacking linguistic diversity and genuine alignment with human design intent.

Inspired by recent benchmarks that prioritize quality over quantity—such as HumanEval (Chen et al.), which uses 164 hand-crafted programming problems to evaluate code

generation, MATH-500 (Hendrycks et al.), which employs expert-written problems to probe mathematical reasoning boundaries, and GAIA (Mialon et al.). Rather than scaling to hundreds of thousands of template-generated examples, we invest in 600 human-verified instances that systematically cover the geometric complexity spectrum (L1-L3) and diverse application domains (L4). Each example undergoes rigorous quality assurance to ensure unambiguous ground truth, correct classification, and faithful correspondence between text descriptions and CAD geometry. This design philosophy enables fine-grained diagnosis of model capabilities and limitations, which large-scale but homogeneous benchmarks often obscure.

## 3. Text2CAD-Bench

We present Text2CAD-Bench, a comprehensive benchmark for evaluating text-to-CAD generation capabilities. The benchmark comprises 600 human-curated examples organized into four benchmark levels, with each example paired with dual-style natural language descriptions.

### 3.1. Design Principles

Our benchmark design is guided by three core principles that address limitations identified in existing datasets.

**Unambiguous Ground Truth.** Each text description in our benchmark corresponds to a unique, fully-specified CAD model with all geometric parameters explicitly defined. Both the geometric description and sequence description for the same example describe identical geometry, ensuring that given the same input, only one correct output exists.

**Systematic Complexity Gradation.** Unlike Text2CAD (Khan et al.), whose benchmark levels primarily reflect textual detail (abstract to expert), we stratify examples by geometric complexity.

**Dual Description Styles.** Real users describe designs differently depending on their expertise and intent. We provide two complementary description styles: geometric descriptions that characterize shapes by their appearance and spatial relationships, and sequence descriptions that specify step-by-step construction sequences. This duality enables evaluation of model robustness to varying input formats and provides insights into which description style better facilitates CAD generation.

### 3.2. Benchmark Construction

Figure 1 illustrates our human-in-the-loop construction pipeline, comprising four stages designed to ensure quality, consistency, and coverage.

**Design Specification.** We analyzed the Text2CAD dataset (Khan et al.) to identify core geometric patterns and

coverage gaps. This analysis revealed that existing data predominantly contains sketch-extrude sequences with limited feature diversity. We extended coverage to include advanced CAD operations—chamfer, fillet, sweep, loft, shell, and complex patterns—essential in professional workflows but absent from current benchmarks. Additionally, we surveyed emerging application domains including medical devices, consumer products, and architectural fabrication to ensure real-world relevance.

**Collaborative Model Authoring.** Human designers specify the target geometry and benchmark level, then iteratively develop CadQuery code with AI assistance (Claude, GPT-4) until successful execution. This collaborative process leverages AI efficiency while maintaining human oversight for quality control. We enforce three validity constraints: (1) all geometric parameters must be fully specified; (2) the generated mesh must be valid and watertight; (3) geometric complexity must match the target benchmark level.

**Description Generation.** Each validated CAD model receives two natural language descriptions following carefully designed templates.

- **Geometric Descriptions.** We adopt a structured description pattern following the sequence: global shape → local features → fine details → global summary. Through preliminary experiments comparing multiple description strategies, we found this pattern yields superior generation results compared to alternatives such as purely sequential descriptions or detail-first approaches. A representative example:

  *"A rectangular enclosure with rounded vertical edges. The top face features a circular opening positioned near one end. All vertical edges are finished with 2mm fillets. The circular opening has a diameter of 15mm, centered 20mm from the shorter edge. The overall dimensions are 80mm in length, 50mm in width, and 30mm in height."*

- **Sequence Descriptions.** We extend the expert-level instruction sequence format introduced in Text2CAD (Khan et al.). Since the original format covers only sketch-extrude operations, we design analogous command-style descriptions for advanced features by referencing standard CAD command conventions, preserving the sequential nature while expanding feature coverage:

  *"Create a workplane on the XY plane. Sketch a rectangle with dimensions 80mm × 50mm and corner radius 5mm. Extrude the sketch 30mm in the positive Z direction. Select the top face and create a new sketch. Draw a circle with diameter 15mm centered at offset (20mm, 25mm) from the face origin. Perform a cut-through*

*operation. Select all vertical edges and apply a fillet with radius 2mm."*

### 3.3. Benchmark Levels

We define four benchmark levels to enable systematic evaluation across the spectrum of text-to-CAD challenges. Levels are determined by geometric complexity and feature distribution, with manual verification as a secondary check. Detailed classification criteria are provided in Appendix A. L1–L3 form a hierarchy based on geometric complexity, while L4 targets real-world application diversity.

#### 3.3.1. GEOMETRIC COMPLEXITY HIERARCHY

The Benchmark levels from L1 to L3 are stratified by the complexity of required CAD operations and geometric features, as summarized in AppendixDetails of Text2CAD-Bench.The progression from L1 to L3 reflects increasing demands on the model's geometric reasoning capabilities:

**L1 (Basic)** tests fundamental understanding of primitive shapes and basic spatial relationships. Success at this level indicates the model can parse geometric descriptions and generate valid, simple CadQuery code.

**L2 (intermediate)** introduces compositional complexity through boolean operations and standard CAD features. Models must correctly sequence multiple operations and handle feature interactions such as applying fillets to edges created by boolean cuts.

**L3 (Advanced)** requires mastery of sophisticated CAD operations that many current models struggle with. Sweep and loft operations demand understanding of path-based geometry.

Basic finishing features such as chamfer and fillet appear across all levels, applied to geometries of corresponding complexity. Complex patterns and freeform surfaces are concentrated in L3 due to their inherent difficulty.

#### 3.3.2. REAL-WORLD APPLICATION SCENARIOS

L4 comprises 100 examples representing real-world design scenarios across diverse application domains. Unlike L1–L3, which focus on isolated geometric capabilities, L4 evaluates whether models can handle the contextual variety of practical text-to-CAD applications. We emphasize that L4 examples still provide precise geometric descriptions with fully specified parameters—they are not underspecified functional requirements. The distinction from L1–L3 lies in:

**Domain diversity:** Examples span industrial/mechanical parts ($\sim$40%), consumer products ($\sim$25%), medical devices ($\sim$15%), architectural elements ($\sim$10%), and educational models ($\sim$10%).

**Contextual richness:** Descriptions may reference application context or domain-specific conventions (e.g., "suitable for M4 screw") that models must interpret correctly.

**Practical relevance:** Examples reflect actual design tasks that may be encountered in emerging text-to-CAD applications rather than abstract geometric exercises.

The geometric complexity of L4 examples varies—some may be simpler than typical L3 examples, while others are comparably complex. L4's primary purpose is assessing generalization across application domains rather than pushing geometric difficulty limits.

### 3.3.3. EVALUATION METRICS

We adopt three complementary metrics to comprehensively assess text-to-CAD generation quality, measuring both code validity and geometric fidelity.

**Chamfer Distance (CD).** Chamfer Distance quantifies the geometric similarity between the generated mesh $M_g$ and the ground-truth mesh $M^*$ via bidirectional nearest-neighbor distances between point clouds. Let $P_g$ and $P^*$ denote point sets uniformly sampled from $M_g$ and $M^*$ respectively:

$$\text{CD}(P_g, P) = \frac{1}{|P_g|} \sum_p d(p, P) + \frac{1}{|P^*|} \sum_q d(q, P_g) \quad (1)$$

We sample 30,000 points per mesh and normalize all models to unit bounding boxes before computation. CD values are reported as $\times 10^3$ for readability. Lower values indicate better geometric fidelity.

**Invalidity Rate (IR).** Invalidity Rate measures the percentage of generated code samples that fail to produce valid geometry. A sample is considered invalid if: the code fails to execute due to syntax or runtime errors,execution exceeds the 60-second timeout, or the output produces empty or degenerate geometry. Lower IR indicates more reliable code generation. This metric captures whether models can produce syntactically correct and executable CadQuery code, complementing geometric fidelity metrics.

**Intersection over Union (IoU).** IoU measures volumetric overlap between the generated and ground-truth models. We voxelize both meshes at resolution $256^3$ and compute:

$$\text{IoU} = \frac{|V_g \cap V^*|}{|V_g \cup V^*|} \quad (2)$$

where $V_g$ and $V^*$ denote the voxelized representations. IoU captures global shape agreement and is less sensitive to surface sampling noise than CD. Higher values indicate better geometric accuracy.

## 4. Experiments

We conduct comprehensive experiments on Text2CAD-Bench to evaluate state-of-the-art general-purpose large language models and domain-specific models. Our evaluation aims to address three key questions: (1) The current capability of LLMs for text-to-CAD generation.(2) Performance degradation with increasing geometric complexity.(3) Prompt style—geometric or sequential is more effective.

### 4.1. Experimental Setup

#### 4.1.1. MODELS

We evaluate two categories of models: general-purpose LLMs and domain-specific models. General-purpose LLMs. We select seven leading large language models that demonstrate strong performance across diverse tasks: **GPT-5.2**(OpenAI et al.), **Claude-4.5-Sonnet**(Anthropic, 2026), **Gemini-3-Flash**(Team et al., 2023), **DeepSeek-V3.2**(DeepSeek-AI et al.) and **Qwen3-max**(Yang et al., 2025).

Domain-specific models. We evaluate three models specifically designed or fine-tuned for CAD generation: **Text2CAD**(Khan et al.), **Text2CADQuery**(Xie & Ju), and **CADFusion**(Wang et al., 2025).

For large language models, we use official APIs without any hyperparameter tuning to ensure fairness. For smaller domain-specific models, we maintain the default settings from their official GitHub repositories or Hugging Face releases, and provide environments and configurations as close as possible to their original states.

#### 4.1.2. IMPLEMENTATION DETAILS

Generated CadQuery code is executed in an isolated Python 3.10 environment with CadQuery 2.4. Successfully executed code produces STEP files, which are converted to point clouds for evaluation using Trimesh. We uniformly sample 30,000 points from each mesh surface. For IoU computation, the voxel resolution is 256. All models are normalized to unit bounding boxes before computing Chamfer Distance.For invalid samples, we do not calculate their CD and IOU.

For the main experiments, we employ a zero-shot prompt template that provides clear instructions for CadQuery code generation. The complete template is provided in Appendix B.

### 4.2. Main Results

Table 1 presents the main results across all models and tires1 to 3. We report CD, IR and IOU for each experiments, along with averages weighted by sample count.

*Table 1.* Main results on Text2CAD-Bench. CD: Chamfer Distance ($\times 10^3$, ↓), IR: Invalidity Rate (%, ↓), IoU: Intersection over Union (↑). Best results are **bolded**

| 3*Model | Geometric Prompt (Geo) | | | | | | | | | Sequence Prompt (Seq) | | | | | | | | |
|---|---|---|---|---|---|---|---|---|---|---|---|---|---|---|---|---|---|---|
| | | L1 | | | L2 | | | L3 | | | L1 | | | L2 | | | L3 | |
| | CD↓ | IR↓ | IoU↑ | CD↓ | IR↓ | IoU↑ | CD↓ | IR↓ | IoU↑ | CD↓ | IR↓ | IoU↑ | CD↓ | IR↓ | IoU↑ | CD↓ | IR↓ | IoU↑ |
| *General-purpose LLMs* | | | | | | | | | | | | | | | | | | |
| GPT-5.2 | **44.31** | 11.1% | **0.59** | **60.38** | 20.0% | **0.50** | 93.46 | 68.0% | 0.23 | **48.73** | 19.5% | **0.62** | **78.21** | 31.6% | **0.47** | 82.94 | 74.0% | 0.25 |
| Claude-4.5-Sonnet | 52.62 | 20.3% | 0.54 | 71.66 | 43.7% | 0.48 | **70.13** | 70.0% | 0.25 | 57.08 | 25.3% | 0.57 | 75.65 | 53.5% | 0.46 | 84.82 | 75.0% | 0.23 |
| DeepSeek-V3.2 | 53.15 | 13.3% | 0.54 | 79.03 | 22.3% | 0.42 | 101.23 | 69.0% | 0.17 | 55.55 | 20.0% | 0.55 | 97.11 | 32.5% | 0.37 | 91.55 | 73.0% | 0.16 |
| Qwen3-max | 84.54 | 21.4% | 0.40 | 107.40 | 33.4% | 0.29 | 148.58 | 92.0% | 0.07 | 68.02 | 30.1% | 0.48 | 113.09 | 49.1% | 0.32 | 97.42 | 93.0% | 0.12 |
| MiniMax M2.11 | 65.92 | 24.8% | 0.45 | 83.06 | 33.4% | 0.40 | 114.33 | 91.0% | 0.26 | 63.63 | 33.3% | 0.46 | 106.07 | 43.7% | 0.32 | **79.84** | 89.0% | **0.28** |
| GLM 4.7 | 67.50 | 13.7% | 0.44 | 88.57 | 26.3% | 0.35 | 92.70 | 83.0% | 0.25 | 68.24 | 20.4% | 0.45 | 100.59 | 34.3% | 0.33 | 107.32 | 84.0% | 0.20 |
| Gemini3-Flash | 66.82 | 17.7% | 0.44 | 88.57 | 12.8% | 0.35 | 91.61 | 12.8% | 0.20 | 105.5 | 16.8% | 0.31 | 105.47 | 16.8% | 0.33 | 108.32 | 45.0% | 0.24 |
| *Domain-specific Models* | | | | | | | | | | | | | | | | | | |
| Text2CAD | 219.57 | **11.0%** | 0.08 | 260.92 | **6.0%** | 0.04 | 234.00 | **2.0%** | 0.04 | 249.68 | **8.0%** | 0.05 | 262.73 | **5.0%** | 0.04 | 266.84 | **6.0%** | 0.03 |
| CADFusion | 229.62 | 60.0% | 0.03 | 236.63 | 70.0% | 0.03 | 209.79 | 53.0% | 0.03 | 212.17 | 65.0% | 0.05 | 242.68 | 64.0% | 0.03 | 191.48 | 51.0% | 0.03 |
| Text2CADQuery | 227.46 | 44.0% | 0.07 | 269.11 | 67.0% | 0.04 | 255.42 | 80.0% | 0.02 | 229.42 | 29.0% | 0.07 | 277.28 | 49.0% | 0.03 | 240.70 | 90.0% | 0.01 |

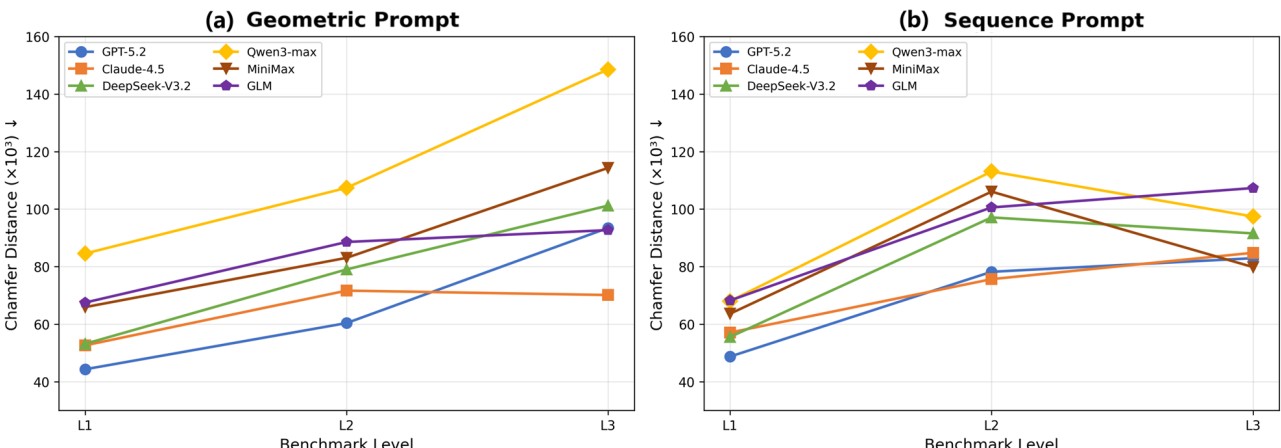

*Figure 2.* Chamfer Distance across benchmark levels for (a) geometric and (b) sequence prompts. General-purpose LLMs consistently outperform domain-specific models in geometric fidelity, while all models exhibit substantial degradation from L1 to L3.

### 4.2.1. OVERALL PERFORMANCE

Table 1 presents comprehensive evaluation results across all models and benchmark levels. Several key findings emerge from the analysis.Figure 2 visualizes the Chamfer Distance across benchmark levels and

Among general-purpose LLMs, GPT-5.2 achieves the best performance on L1-L2 tasks with the lowest Chamfer Distance (44.31 on L1-Geo) and highest IoU of 0.59. Claude-4.5-Sonnet and DeepSeek-V3.2 form a competitive second tier, while Qwen3-max exhibits weaker geometric reasoning, particularly on complex tasks. Domain-specific models yield substantially higher CD despite lower Invalidity Rates—a finding we discuss below.

Performance Degradation. All models exhibit substantial degradation from L1 to L3. Chamfer Distance increases by 1.3–2.1× on average, while Invalidity Rate rises from approximately 15% to 70–90%. The steepest decline occurs between L2 and L3, where advanced features such as sweep, loft, and shell operations cause execution rates

to plummet. Notably, Claude-4.5-Sonnet demonstrates the most graceful degradation, achieving the lowest L3-Geo CD (70.13)—surpassing GPT-5.2 (93.46) despite trailing on simpler tasks.

Prompt Style Comparison. Geometric descriptions outperform sequence descriptions on L1-L2, where simple shapes can be concisely characterized by visual appearance. However, this advantage reverses on L3: GPT-5.2 (82.94 vs. 93.46), DeepSeek-V3.2 (91.55 vs. 101.23), and MiniMax (79.84 vs. 114.33) all perform better with sequence prompts. We attribute this to the procedural nature of advanced CAD operations—appearance-based descriptions struggle to convey complex constructive geometry concisely, whereas sequence descriptions directly encode construction logic in a structured format. Further analysis in Appendix A corroborates this finding.

General-purpose vs. Domain-specific Models. A striking divergence emerges between model categories. Text2CAD achieves the lowest IR (as low as 2%) yet the highest CD ( 220), indicating it generates executable code without cap-

Table 2. L4 evaluation results by domain category. Geo: Geometry Score, Task: Task Completion Score (scale: 1–10, ↑). Non-Industrial includes consumer products, medical devices, architectural elements, and educational models. Best results are **bolded**.

| Model | Industrial | | Non-Industrial | | Avg. |
|---|---|---|---|---|---|
| | Geo | Task | Geo | Task | |
| *General-purpose LLMs* | | | | | |
| GPT-5.2 | 1.82 | 1.65 | 1.54 | 1.40 | 1.60 |
| Claude-4.5 | 1.54 | 1.42 | 1.30 | 1.18 | 1.36 |
| DeepSeek-V3.2 | 1.45 | 1.32 | 1.19 | 1.07 | 1.26 |
| Qwen3-max | 4.25 | 3.88 | 3.63 | 3.29 | 3.76 |
| MiniMax | **5.12** | **4.68** | **4.48** | **4.06** | **4.59** |
| GLM | 4.08 | 3.72 | 3.48 | 3.14 | 3.61 |
| *Domain-specific Models* | | | | | |
| Text2CAD | 2.45 | 2.12 | 2.02 | 1.71 | 2.08 |
| CADFusion | 4.15 | 3.82 | 3.66 | 3.32 | 3.74 |
| Text2CADQuery | 3.82 | 3.48 | 3.29 | 2.97 | 3.39 |

turing geometric specifications. This highlights the importance of multi-metric evaluation: benchmarks relying solely on execution rate risk rewarding geometrically incorrect outputs.

### 4.3. Real-World Application Scenarios

Unlike L1-L3, which test isolated geometric capabilities, L4 evaluates model performance on realistic design tasks drawn from diverse fields. These results highlight the importance of domain diversity in text-to-CAD benchmarks. Models that perform well on standard mechanical parts may struggle with emerging applications in medical device design or consumer product customization, where geometric conventions and description styles differ from traditional CAD workflows.

### 4.4. Ablation Studies

To validate our choice of output representation, we compare CadQuery against DeepCAD-style command sequences on L1 and L2 subsets, which provide sufficient diversity while maintaining experimental tractability. For command sequence generation, we adopt PythonOCC library built on the OpenSCAD kernel—the same backend that DeepCAD's domain-specific language (DSL) parses into for execution.

Table 3 presents the results. Under command sequence representation, all metrics exhibit varying degrees of degradation compared to CadQuery (Table 1). Qwen3-max shows particularly severe performance drops (CD increases from 49.85 to 168.64 on L1), likely due to the scarcity of similar tasks in its pretraining corpus. Gemini-3-Flash demonstrates

Table 3. Output representation comparison: Command Sequence results on L1-L2.

| Model | Prompt | CD↓ | IoU↑ | IR↓ |
|---|---|---|---|---|
| *L1* | | | | |
| Gemini-3-Flash | Geo | 64.26 | 0.51 | 16.8% |
| | Seq | 52.62 | 0.56 | 21.4% |
| Qwen3-max | Geo | 168.64 | 0.11 | 49.5% |
| | Seq | 164.23 | 0.16 | 52.7% |
| *L2* | | | | |
| Gemini-3-Flash | Geo | 194.14 | 0.12 | 29.8% |
| | Seq | 193.45 | 0.11 | 25.5% |
| Qwen3-max | Geo | 219.47 | 0.04 | 71.6% |
| | Seq | 232.87 | 0.05 | 79.3% |

more moderate degradation but still underperforms its CadQuery counterpart across all settings. Notably, even when using sequence-style prompts that align with command sequence syntax, performance fails to match CadQuery results—on L2, Gemini achieves CD of 193.45 with sequence prompts versus 48.92 with CadQuery, representing a 4× degradation. The L2 results reveal comprehensive deterioration across all metrics, suggesting that command sequences struggle particularly with compositional geometric operations. These findings confirm that code-based representations like CadQuery better leverage LLMs' pretrained capabilities, supporting the recent trend toward Python-based CAD generation in the research community.

### 4.5. Qualitative Results

Figure 3 presents qualitative comparisons across representative samples. Several failure patterns emerge. First, even basic L1 tasks cause execution failures for multiple models (MiniMax, Qwen, Gemini), indicating fundamental issues with CadQuery API usage. Second, successful execution does not guarantee geometric correctness. Third, prompt style affects generation differently across models: GPT fails on L1_109 with geometric prompts but succeeds with sequence prompts, while Claude shows the opposite pattern. These observations complement our quantitative metrics and demonstrate that comprehensive evaluation requires both execution validity and geometric fidelity assessment.

## 5. Discussion & Conclusion

We presented Text2CAD-Bench, a comprehensive benchmark for evaluating text-to-CAD generation capabilities. Through extensive experiments on general-purpose LLMs and domain-specific systems, we arrive at three key conclusions.

**Text-to-CAD remains largely unsolved beyond basic ge-**

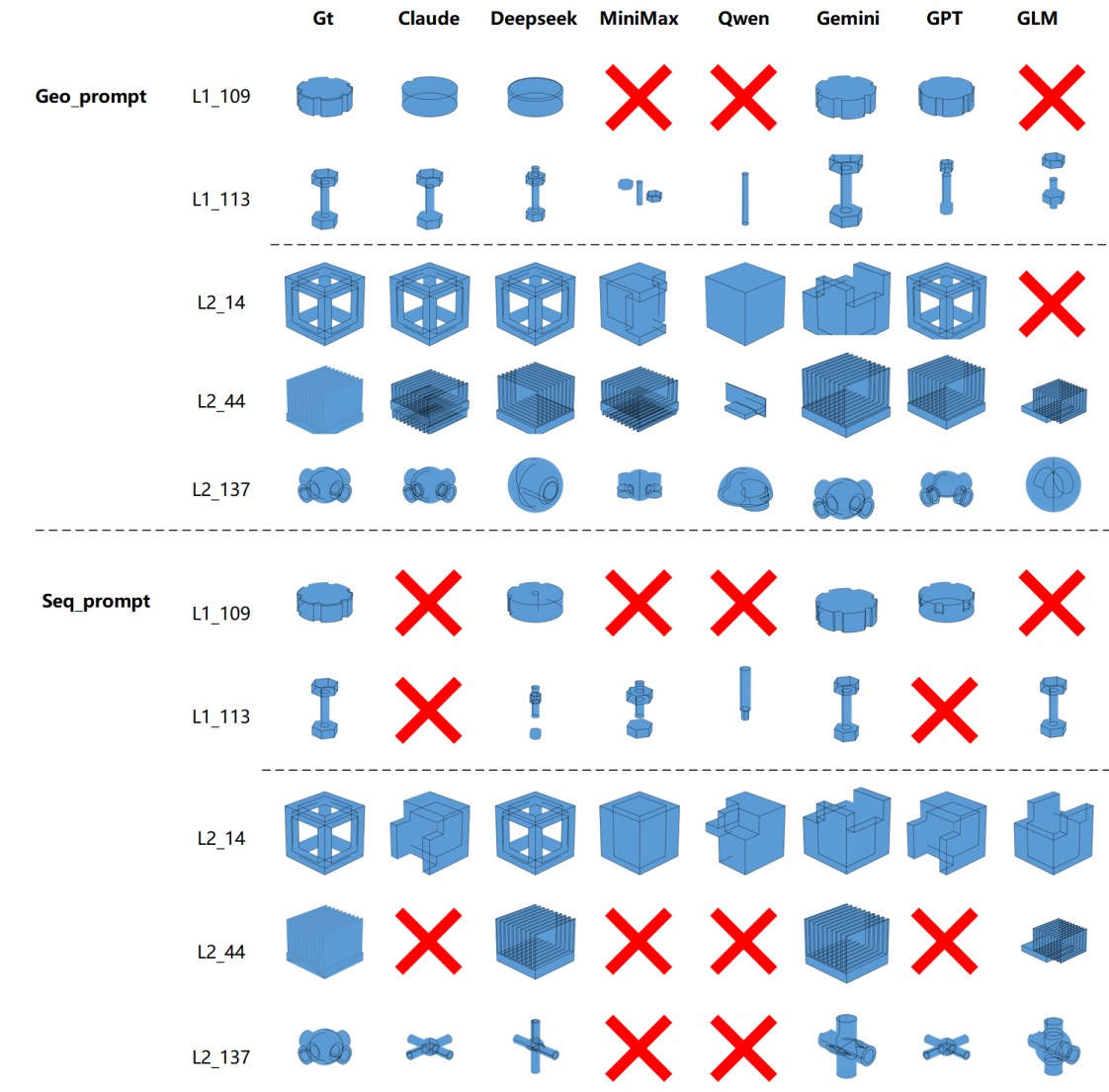

*Figure 3.* Qualitative comparison of generated CAD models

**ometry.** While current models achieve reasonable performance on L1-L2 tasks, performance degrades substantially on L3 advanced features. Significant advances in model capabilities or training data are needed before text-to-CAD can reliably support professional workflows.

**Output representation matters critically.** Our comparison demonstrates that Python-based CadQuery representations enable models to exploit pretrained code generation capabilities, yielding substantial improvements over command sequences in both execution validity and geometric accuracy.

**Prompt style influences generation quality.** Geometric descriptions yield better results on L1-L2 tasks, while sequence descriptions show advantages on L3 advanced features. The "global→local→detail→global" pattern aligns well with how language models process spatial information.

**Limitations.** While Text2CAD-Bench covers broader geometric complexity and application domains than existing benchmarks, 600 examples cannot exhaustively represent the full CAD design space. Specialized domains and categories remain underrepresented. We evaluated models without fine-tuning; multi-turn interaction with execution feedback remains unexplored.

**Conclusion.** Text2CAD-Bench provides a challenging testbed for measuring text-to-CAD. Our results establish baselines and reveal advanced CAD features as a significant capability frontier. Part of the benchmark, evaluation code, are available at https://huggingface.co/datasets/AICAD/Text2CAD-Bench.

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

# A. Details of Text2CAD-Bench

## A.1. Dataset Statistics

Table 4 summarizes the key statistics of Text2CAD-Bench across complexity levels. We report the average word count for both description styles, average lines of CadQuery code, and average number of API calls in the ground-truth implementations.

*Table 4.* Statistics of Text2CAD-Bench across complexity levels. We report mean values for description length (word count), code complexity (lines of code), and API usage (number of CadQuery API calls).

| Level | Geo Words | Seq Words | Code Lines | API Calls |
|---|---|---|---|---|
| L1 (Basic) | 89.9 | 81.5 | 7.9 | 10.8 |
| L2 (Intermediate) | 108.4 | 109.6 | 19.1 | 15.0 |
| L3 (Advanced) | 429.4 | 173.6 | 70.7 | 26.8 |

Several observations emerge from these statistics:

**Complexity gradation is reflected in code structure.** The average lines of code increase substantially from L1 to L3 (7.9 $\rightarrow$ 19.1 $\rightarrow$ 70.7), representing an approximately 9× increase from basic to advanced levels. Similarly, API call counts grow from 10.8 to 26.8, indicating that L3 examples require more sophisticated operation sequences. This progression validates our complexity stratification criteria.

**Description styles exhibit different scaling patterns.** While both geometric and sequence descriptions grow longer with complexity, geometric descriptions scale more dramatically at L3 (429.4 words) compared to sequence descriptions (173.6 words). This divergence reflects the inherent challenge of describing complex freeform geometry in natural language: intricate shapes such as swept surfaces or lofted profiles require extensive spatial elaboration, whereas procedural sequences maintain relative compactness through structured command syntax.

**L1-L2 descriptions are comparable in length.** At lower complexity levels, geometric and sequence descriptions have similar word counts, suggesting that simple geometries can be described with comparable verbosity in either style. The divergence at L3 indicates that description style choice becomes more consequential as geometric complexity increases.

# B. Limitations

While Text2CAD-Bench represents a significant step toward comprehensive evaluation of text-to-CAD generation, we acknowledge several limitations that suggest directions for future work.

**Benchmark Scope**

**Domain coverage.** The application domains included in L4 (industrial/mechanical, consumer products, medical devices, architectural elements, and educational models) were selected based on the authors' assessment of fields where CAD-based manufacturing is prevalent, valuable, or has emerging demand. This selection is inherently non-exhaustive. Specialized domains such as aerospace, automotive, jewelry design, and microelectronics—each with distinct geometric conventions and precision requirements—remain underrepresented. Future benchmark extensions should incorporate domain experts to ensure broader and more representative coverage.

**Geometric categories.** Our benchmark focuses primarily on solid modeling operations. Important CAD categories such as sheet metal design, multi-part assemblies, and parametric families with complex constraints are not covered. These categories present unique challenges (e.g., bend allowance calculations, mate constraints) that warrant dedicated evaluation.

**Sample size.** With 600 examples, Text2CAD-Bench cannot exhaustively represent the full space of CAD design tasks. While we prioritize quality and diversity over quantity, certain geometric configurations and edge cases may be underrepresented.

**Representation Choice**

We exclusively adopt CadQuery as our target representation. While this choice is well-motivated (Section 2.2), it limits direct comparison with systems targeting other CAD formats such as OpenSCAD, FreeCAD scripts, or proprietary formats (e.g., SolidWorks macros, AutoCAD LISP). Models optimized for alternative representations may exhibit different capability

*Table 5.* Statistics of Text2CAD-Bench.

| Statistic | Value |
|---|---|
| Total examples | 600 |
| L1 | 200 |
| L2 | 200 |
| L3 | 100 |
| L4 | 100 |
| Total prompts | 1,200 |
| Geometric descriptions | 600 |
| Sequence descriptions | 600 |
| **Application domains (L4)** | |
| Industrial/Mechanical | $\sim$40% |
| Consumer Products | $\sim$25% |
| Medical Devices | $\sim$15% |
| Architectural Elements | $\sim$10% |
| Educational Models | $\sim$10% |

profiles that our benchmark does not capture.

**Evaluation Protocol**

We evaluated models exclusively in zero-shot and few-shot settings without fine-tuning. Models specifically trained on our benchmark distribution might achieve substantially different results. The relative ranking of models could shift significantly under fine-tuning regimes.Besides,We did not explore multi-turn interaction, where iterative refinement based on execution feedback or visual inspection could potentially improve generation quality. Such interactive workflows may better reflect practical text-to-CAD usage patterns.

The rapid pace of LLM development means that evaluation results can quickly become outdated. New models with improved code generation capabilities are released frequently, and existing models receive continuous updates.To address this challenge, **we establish a public leaderboard** alongside our benchmark release. We encourage researchers and practitioners to evaluate their models on Text2CAD-Bench and submit results to the leaderboard, fostering continuous benchmarking and community engagement. The leaderboard will be maintained at [URL] and accepts submissions following the standardized evaluation protocol described in Section 4.1.

# C. Prompts Used in Text2CAD-Bench

This appendix provides the complete prompt templates used in our evaluation. All experiments employ a two-stage prompting strategy: an initial generation prompt and, upon execution failure, a retry prompt that incorporates error feedback.

## C.1. System Prompt

The system prompt establishes the model's role as a CadQuery expert and provides critical API guidance to prevent common errors. We include five diverse examples spanning different geometric operations (chamfer, sweep, loft, twist extrude, and fillet) to demonstrate proper CadQuery usage patterns.

For few-shot experiments, we augment the base prompt with additional input-output demonstrations. These examples are drawn from a held-out set that shares the same data source as the evaluation benchmark but contains no overlapping samples. The selected demonstrations cover diverse advanced geometric features (e.g., sweep, loft, shell, complex boolean operations), enabling models to learn correct API usage patterns for operations that may be underrepresented in their pretraining corpora. Specifically, we prepend $k$ demonstration pairs before the target task description, where each pair consists of a natural language description and its corresponding ground-truth CadQuery implementation.

```
You are an expert CadQuery programmer. Generate Python code
using CadQuery to create 3D models.
```

```
CRITICAL RULES:
1. MUST start with: import cadquery as cq
2. MUST assign final result to variable named 'result'
3. DO NOT use show_object(), display(), or any visualization
4. DO NOT include explanations, ONLY code in '''python''' block

CADQUERY API NOTES (IMPORTANT - avoid common errors):
- There is NO .cone() method. For cones, use:
  cq.Solid.makeCone(radius1, radius2, height)
- There is NO .array() method. Use
  .rarray(xSpacing, ySpacing, xCount, yCount) for rectangular arrays
- For circular patterns, use
  .polarArray(radius, startAngle, angle, count)
- .fillet() and .chamfer() require a solid first.
  Always create geometry before filleting.
- Use .box(length, width, height) for boxes
- Use .cylinder(height, radius) for cylinders
- Use .sphere(radius) for spheres
- Chain operations properly:
  cq.Workplane("XY").box(10,10,10).faces(">Z").hole(2)
- For extrusion: .extrude(height)
- For cutting: .cut(other_solid)
- For union: .union(other_solid)

[Five in-context examples omitted for brevity;
see supplementary materials for complete prompt]

OUTPUT FORMAT:
'''python
import cadquery as cq
# your code here
result = ...  # final CadQuery Workplane or Shape
'''
```

### C.1.1. IN-CONTEXT EXAMPLES

The system prompt includes five carefully designed examples that demonstrate diverse CadQuery operations:

**Chamfered Hexagonal Prism**. Demonstrates polygon creation and edge chamfering with face selection.

**Helical Spring**. Demonstrates sweep operation along a programmatically generated helix path.

**Circle to Square Loft**. Demonstrates loft operation between dissimilar cross-sections.

**Twisted Cycloid Gear**. Demonstrates parametric curves and twist extrusion for complex geometry.

**L-Bracket with Fillet and Chamfer**. Demonstrates profile sketching and combined fillet/chamfer operations with edge selection.

These examples were selected to cover operations frequently required in L2-L3 tasks while avoiding overlap with evaluation samples. For each evaluation sample, the user prompt simply provides the task description:

```
{description}
```

where {description} is replaced with either the geometric or sequence description from our benchmark.

### C.2. Retry Prompt

When generated code fails to execute, we provide the model with error feedback and request a corrected version. This retry mechanism allows models to leverage error messages for self-correction.

```
The previous CadQuery code attempt failed with an error.
Please analyze the error and generate corrected code.

## Original Task
{original_prompt}
```

```
## Previous Code Attempt (Iteration {iteration}):
```python
{previous_code}
```

## Error Message:
{error_message}

## Instructions:
1. Carefully analyze the error message above
2. Identify the root cause of the failure
3. Generate corrected CadQuery code that fixes the issue
4. Ensure the code follows all CadQuery API rules
   mentioned in the system prompt
5. Output ONLY the corrected code in a ```python``` block

Please generate the corrected code:
```

## D. Details in Building Text2CAD-Bench

The construction of Text2CAD-Bench follows a rigorous human-in-the-loop pipeline designed to ensure quality, consistency, and geometric diversity. This appendix provides additional details on our collaborative authoring process and quality control procedures.

Our benchmark construction involved a team of three contributors with backgrounds in mechanical engineering and CAD modeling. The construction process spanned approximately four weeks, with each example undergoing multiple iterations before acceptance into the final benchmark.

For each example, the authoring process begins with a human designer specifying the target geometry, intended difficulty level, and key geometric features to be incorporated. The designer first composes the natural language descriptions—both geometric and sequence styles—that fully specify all parameters of the intended model. This description-first approach ensures that the textual specification serves as the authoritative reference, preventing post-hoc rationalization of implementation details.

Following description authoring, the designer develops the corresponding CadQuery implementation, often consulting AI assistants (Claude, Gemini and GPT) for API syntax, debugging assistance, or alternative implementation strategies. This collaborative approach leverages AI efficiency for routine coding tasks while maintaining human oversight for geometric correctness and difficulty calibration. Importantly, AI assistants serve as coding aids rather than autonomous generators—all geometric decisions and final code validation remain under human control.

For geometric descriptions, authors compose natural language following the "global shape → local features → fine details → global summary" pattern established in Section 3.2. For sequence descriptions, we adopt a simplified specification format compared to verbose coordinate-based representations. For instance, rather than specifying a rectangle by its four corner coordinates (e.g., "draw lines from (0,0) to (80,0) to (80,50) to (0,50) and back to origin"), we describe it by center point, orientation, and dimensions (e.g., "In XY plant, sketch a rectangle centered at the origin with length 80mm and width 50mm"). This simplification reduces description verbosity while preserving complete geometric specification, and better reflects how practitioners naturally communicate design intent.

The iterative refinement process typically requires 2–5 iterations per example. Common revision triggers include: execution errors due to API misuse, geometric output deviating from design intent, code complexity misaligned with target difficulty level, and insufficient feature coverage for the designated tier. Each iteration involves code modification, execution validation, and visual inspection of the generated mesh against the intended geometry.An independent reviewer (not the original author) verifies each example against four criteria: (1) code executes without errors or warnings in the standardized environment; (2) generated geometry accurately reflects both description styles; (3) geometric complexity aligns with the assigned difficulty level; (4) geometric and sequence descriptions are mutually consistent, describing identical geometry. Examples failing any criterion are returned for revision or excluded from the benchmark.

To ensure consistency across contributors, we established detailed annotation guidelines specifying description conventions, difficulty classification criteria, and common pitfalls to avoid. Regular calibration sessions were conducted to align reviewers' judgments on edge cases, particularly for examples near difficulty level boundaries.

## E. Output Representation of L3 and L4

We provide visualizations of selected L3 and L4 outputs to illustrate characteristic failure modes on advanced features. For each example, ground-truth models are shown alongside generated outputs from representative models.

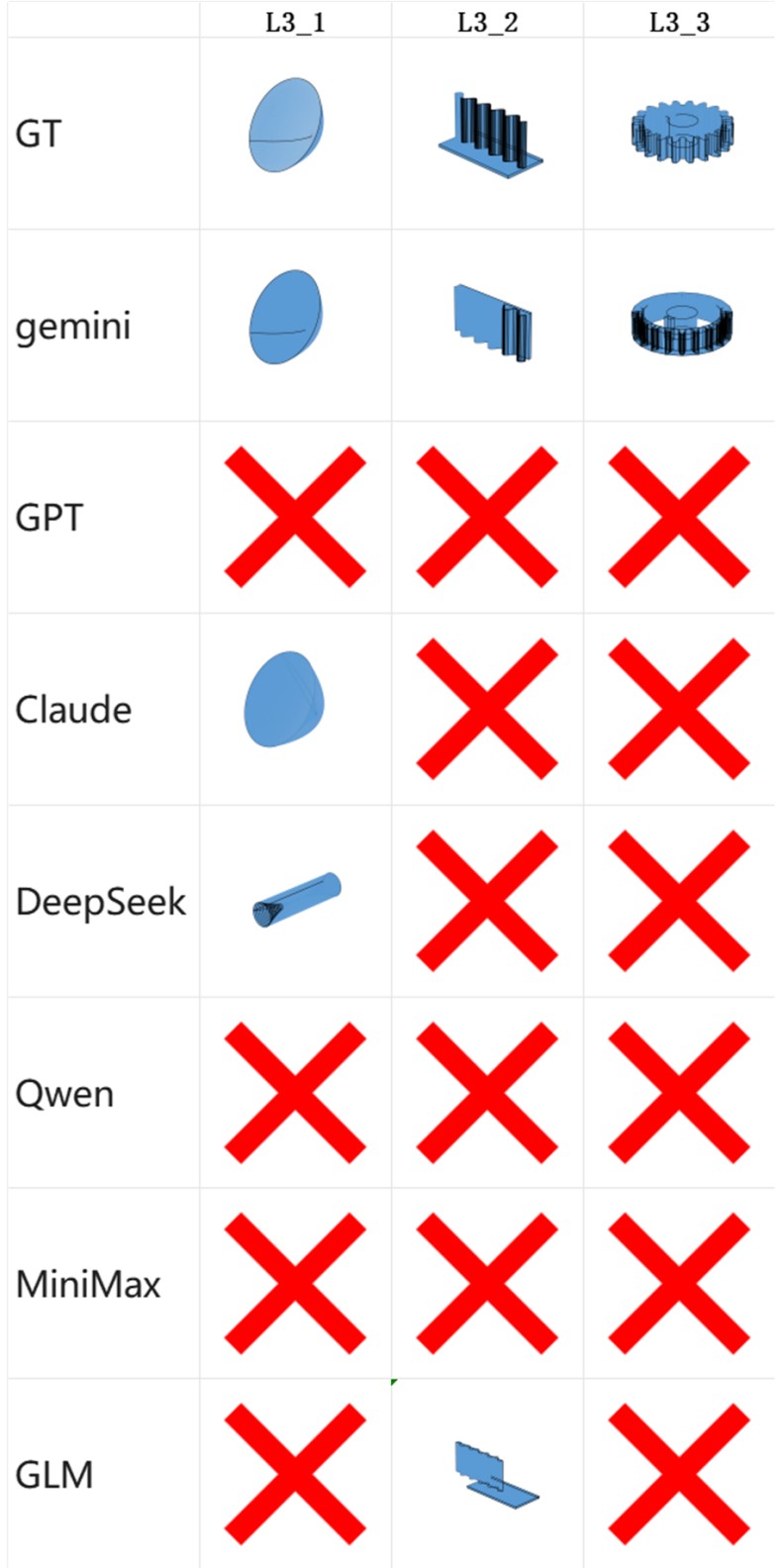

*Figure 4.* Qualitative results on L3 in geometric prompt

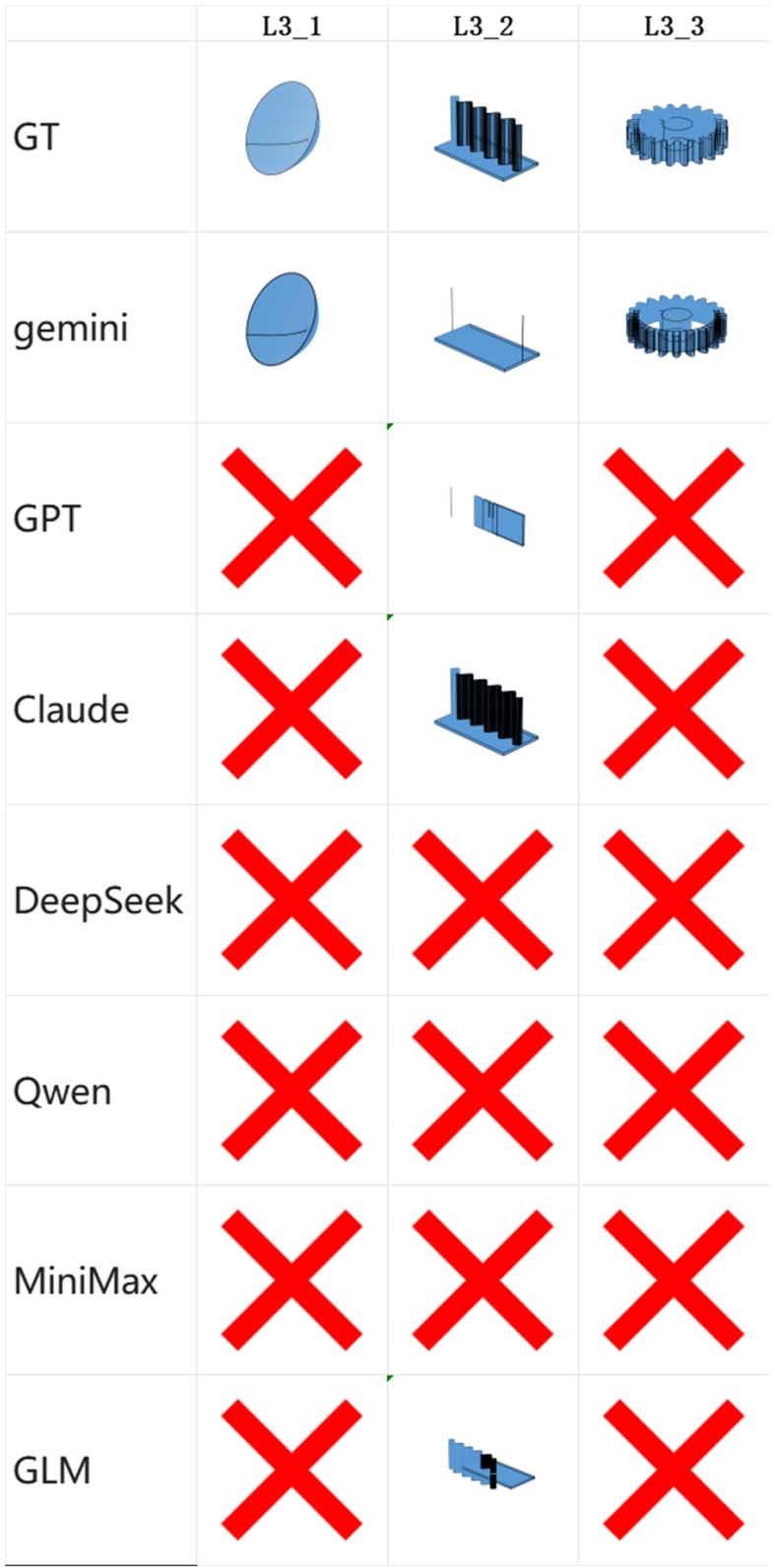

*Figure 5.* Qualitative results on L3 in sequence command

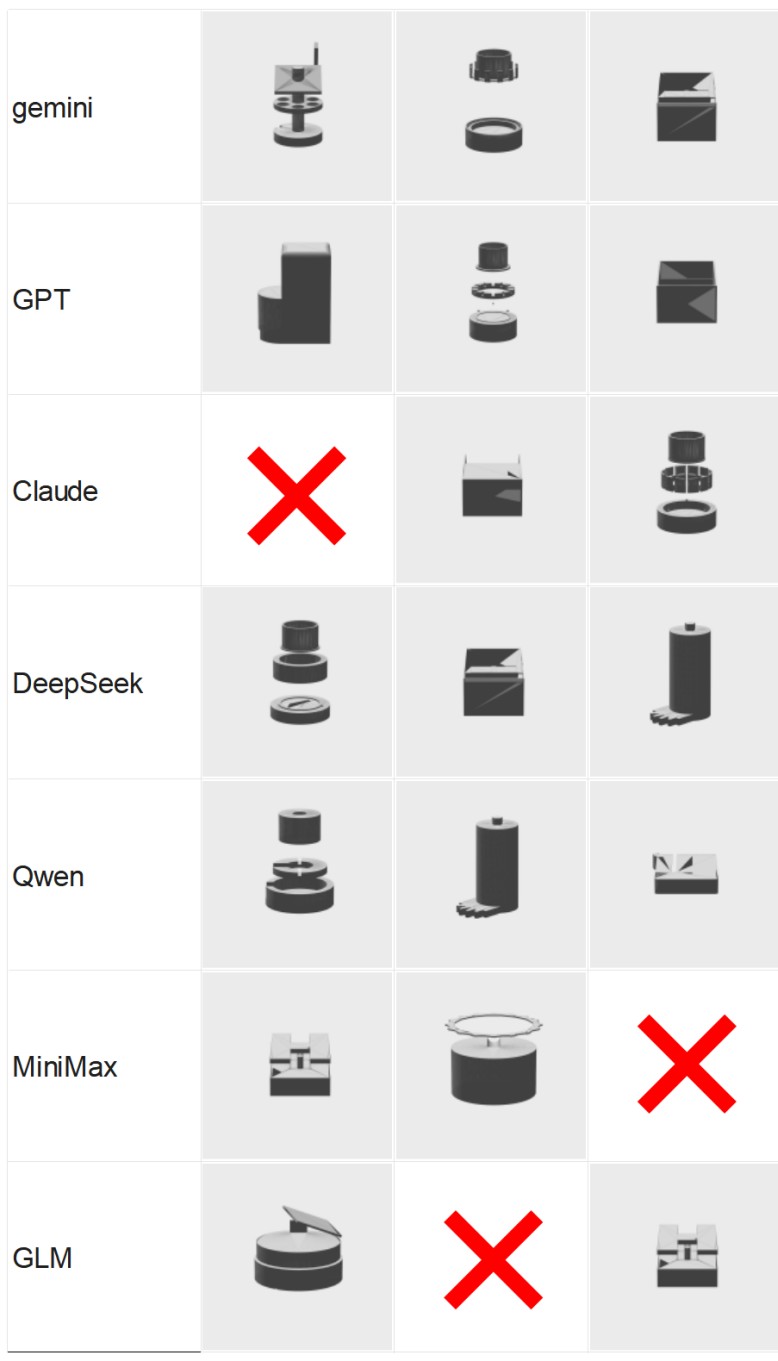

*Figure 6.* Qualitative results on L4 (Real-world) examples across diverse application domains.

