# OpenReview forum: "Text2CAD-Bench: A Benchmark for LLM-based Text-to-Parametric CAD Generation"
_ICML.cc/2026/Conference — Submitted to ICML 2026_

### Official Review · Reviewer_daNp · 2026-03-10

**Soundness:** 2
**Presentation:** 3
**Significance:** 2
**Originality:** 3
**Overall Recommendation:** 3
**Confidence:** 5

**Summary:**

This paper introduces Text2CAD-Bench, a comprehensive benchmark for evaluating LLMs on text-to-parametric CAD generation, which addresses the limitations of existing datasets such as limited operations, narrow domains, and low linguistic diversity.
The authors curated 600 examples across four difficulty levels—ranging from basic primitives (L1) to complex real-world applications (L4), which brings complex operations like sweep and loft.
Another key innovation is the use of dual-style prompts (geometric descriptions and procedural sequences), which aligns with executable CadQuery Python scripts.
Evaluations demonstrate that while general-purpose LLMs outperform domain-specific models in geometric fidelity, all models struggle significantly with complex operations.
Additionally, the study reveals that geometric prompts are superior for simple tasks, whereas procedural sequence prompts are much m

**Compliance With Llm Reviewing Policy:**

Affirmed.

**Final Justification:**

I maintain my score due to the lack of sufficient comprehensiveness as a benchmark, as stated in my reviews.

**Key Questions For Authors:**

1. How is the complexity of path since sweep and loft operation is included?
2. If the authors intend to develop a full-scale dataset based on this benchmark, what would be your proposed methodology?

**Limitations:**

Yes.

**Strengths And Weaknesses:**

**Strengths**
1. The paper fills a critical gap in text-to-CAD evaluation by moving beyond simplistic sketch-extrude datasets to systematically include advanced operations (e.g., lofts, sweeps, fillets, chamfers).
2. The introduction of dual-style prompts (geometric descriptions vs. procedural sequences) is highly insightful. It yields valuable empirical findings, demonstrating that LLMs prefer geometrical descriptions for simple tasks but require explicit procedural logic for complex geometric constructions.
3. The authors' choice to use CadQuery Python scripts over traditional command sequences is well-motivated and supported by convincing ablation studies, proving that code representations might have a better leverage the pre-trained capabilities of LLMs.
4. The number of baseline models is sufficient, and the extensive experimental evaluation is commendable.
5. The human-in-the-loop curation pipeline ensures unambiguous ground truths. The use of multi-metric evaluation (Chamfer Distance, Intersection over Union, and Invalidity Rate) correctly prevents models from being rewarded merely for generating executable but geometrically incorrect code.
7. The writing is clear.

**Weaknesses**
1. While the 'quality over quantity' approach is noted, a benchmark of only 600 examples appears insufficient to represent the true complexity of CAD engineering. Specifically, the current version fails to address multi-part assemblies and inter-component constraints—fundamental capabilities in practical workflows where defining how parts interact and align is crucial. Given this restricted scope, a noticeable dilemma emerges: if the primary goal is to evaluate advanced single-part features (such as 'sweep' or 'loft'), a moderate or large-scale dataset—rather than a small benchmark—would be necessary to provide a solid empirical foundation. Conversely, if the authors intend to offer a highly curated, small-scale benchmark, it must encapsulate high-level engineering challenges, such as part combinations and assembly logic. By delivering neither the statistical scale of a comprehensive dataset nor the assembly-level complexity of an advanced benchmark, the technical depth of this study is noticeably diminished. This ultimately compromises the overall significance and utility of the work for the broader CAD community.
2. What's more, the strict reliance on CadQuery limits the benchmark's ability to directly evaluate text-to-CAD systems optimized for other standard formats (e.g., OpenSCAD, FreeCAD scripts, or proprietary B-rep kernels) without introducing translation layers.
It might be better if this bench could include multiple code representations like OpenSCAD and FreeCAD scripts, which could be adopted by different models trained on different code representation.
3. There should be some examples of prompt-code pair of L1-L4. Otherwise, it remains unclear what the actual code looks like and how the benchmark distinguishes between different difficulty levels and different domain. Providing such details is essential for a comprehensive understanding of the work.
4. the attached huggingface url contains risk of violating the double-blind policy.
5. The parentheses in formula (1) are incorrectly formatted.
6. There is no GT image/shape on Figure 6. Images on page 16 and 17 seems to be the same qualitative results on L3 in geometric prompt with different versions.

---

> ### Author Rebuttal · Authors · 2026-03-31
>
> We thank Reviewer daNp for the thorough and detailed review. We address each concern below.
>
> ## Q1: Benchmark scale (600 examples) is insufficient and lacks multi-part assemblies, creating a dilemma between scale and scope.
>
> Our benchmark already poses substantial challenges: L3 Invalidity Rates reach 70-90% across all general-purpose LLMs, indicating that current models are far from solving even single-part advanced geometry. Scaling up before models can handle existing complexity would dilute diagnostic value. This aligns with established benchmarks — HumanEval (164 problems) and MATH-500 (500 problems) effectively diagnose capability boundaries through curation rather than scale.
> Regarding multi-part assemblies, we respectfully argue that assembly modeling constitutes an independent research dimension involving inter-part constraints, mate relationships, and tolerance specifications — challenges fundamentally different from single-part geometric reasoning. Incorporating assemblies would shift the benchmark's focus and conflate two distinct capability dimensions. We consider assembly-level evaluation an important direction for future work that deserves dedicated benchmark design, rather than being added as an extension to the current framework.
>
> ## Q2: Strict reliance on CadQuery limits models optimized for other formats (OpenSCAD, FreeCAD).
>
> We clarify that Text2CAD-Bench is an end-to-end evaluation: text input, CAD model output. Our metrics (CD, IR, IoU) evaluate the final geometric output, not the intermediate code representation. Any model capable of producing valid geometry from text — regardless of whether it generates CadQuery, OpenSCAD, FreeCAD scripts, or other formats — can be evaluated on our benchmark through the same geometric metrics. In fact, our ablation study (Table 3) demonstrates this flexibility: we evaluated models using DeepCAD-style command sequences (via PythonOCC) alongside CadQuery, comparing outputs through the same CD/IR/IoU metrics. CadQuery serves as our ground-truth representation due to its superior expressiveness, but the evaluation framework itself is representation-agnostic.
>
> ## Q3: No prompt-code pair examples for L1-L4, making it unclear how levels differ.
>
> We acknowledge this gap in presentation. We will include representative prompt-code pairs for each level in our revision, clearly illustrating the progression in geometric complexity, code structure, and description style across L1-L4. These examples will also be made available in our open-source release.
>
> ## Q4: HuggingFace URL risks violating double-blind policy.
>
> We sincerely apologize for this oversight and will ensure full compliance with the double-blind policy in our revision.
>
> ## Q5: Formula (1) formatting errors; Figure 6 lacks ground-truth images.
>
> We will correct the parentheses in Formula (1). Regarding Figure 6, since L4 examples are evaluated through VLM-based scoring (GLM4.6V) rather than geometric comparison with ground truth, Figure 6 is intended to provide qualitative visualization of model outputs across diverse application domains. The evaluation of L4 relies on structured VLM scoring with human verification of score validity, rather than visual GT comparison. We will clarify this in the figure caption.
>
> ## Q6: How is the complexity of paths handled for sweep and loft operations?
>
> For sweep and loft operations in L3, we adopt standard parametric curve formulations (e.g., Bezier curves, helical paths) to define paths, rather than introducing arbitrary complexity. The resulting complexity is validated through our dataset statistics (Appendix A, Table 4): L3 examples average 70.7 lines of code and 26.8 API calls, representing approximately 9x and 2.5x increases over L1, respectively. This substantial growth in code complexity, combined with the 70-90% invalidity rates observed in experiments, confirms that standard parametric paths already pose significant challenges for current models.
>
> ## Q7: If extending to a full-scale dataset, what is the proposed methodology?
>
> We envision systematic expansion through parametric augmentation of existing examples and annotation of real-world engineering designs, while preserving the curation quality that defines our current benchmark. We note that the current 600 examples already reveal substantial capability gaps (70-90% L3 IR), suggesting that addressing these foundational challenges is a productive near-term priority alongside future scaling efforts.

---

> > ### Author Rebuttal · Reviewer_daNp · 2026-04-04
> >
> > I thank authors for their responses. For my concerns:
> >
> > Q1: I'm not convinced. As I said, if paper is to create a **general** text2CAD-bench, assembly modeling should be included and evaluated, otherwise there is a mismatch of the expected scope and the actual coverage.
> >
> > Q2: resolved.
> >
> > Q3: Not conviced. These examples are easy to included in the response instead of waiting until future revision.
> >
> > Q4: resolved.
> >
> > Q5: resolved.
> >
> > Q6: resolved.
> >
> > Q7: not resolved. Similar to Q1, substantial capability gaps revealed by the 600 samples only demonstrate the performance of existing methods, and does not necessarily indicate the comprehensiveness of the benchmark.

---

> > > ### Author Response · Authors · 2026-04-08
> > >
> > > > **Q1 & Q7:** We sincerely appreciate the reviewer's persistence on this point. We acknowledge that the current benchmark does not cover multi-part assemblies, and will add explicit scope discussion and outline a concrete assembly-level extension plan in the revised Conclusion section.
> > > > We believe the single-part focus is justified for two reasons. First, L3 Invalidity Rates of 70-90% across all state-of-the-art models show that even single-part advanced geometry remains a significant unsolved challenge — a focused benchmark enables precise diagnosis of these limitations. Second, assembly modeling involves qualitatively different challenges (inter-part constraints, mate relationships, tolerance specifications) orthogonal to single-part geometric reasoning. Combining both would conflate distinct capability dimensions, similar to how HumanEval and SWE-Bench separately address function-level and repository-level code generation.
> > > > Regarding comprehensiveness (Q7), no model has saturated even L2 performance, confirming that the benchmark provides meaningful discriminative power at the current stage and a solid foundation for future extensions including assembly-level challenges.
> > >
> > > > **Q3:** We provide representative prompt-code pairs for each level:
> > >
> > > > **L1 (Basic):**
> > >
> > > > *Geo prompt:* "The model is a prismatic solid based on a regular hexagon, with uniform material throughout. The base profile is a regular hexagon with an inscribed circle diameter of 50mm, centered at the origin. The hexagonal prism is extruded 20mm in the positive Z-axis direction. At the center of the top surface, there is a circular through-hole with a diameter of 20mm. The six outer edges on the top surface are chamfered at 45 degrees with a chamfer width of 2mm."
> > >
> > > > *Seq prompt:* "Set the XY plane as the base sketch plane. Draw a regular hexagon centered at the origin with an inscribed circle diameter of 50.0mm. Extrude 20.0mm in the +Z direction. On the top face, draw a concentric circle with diameter 50.0mm and perform an extruded cut through all. Apply a 2.0mm chamfer to the six outer edges of the top face."
> > >
> > > > ```python
> > > > from math import cos, pi
> > > > import cadquery as cq
> > > > result = (
> > > >     cq.Workplane("XY")
> > > >     .polygon(6, 50 / cos(pi/6))
> > > >     .extrude(20)
> > > >     .faces(">Z").workplane()
> > > >     .circle(10)
> > > >     .cutThruAll()
> > > >     .faces(">Z").edges("%Line")
> > > >     .chamfer(2)
> > > > )
> > > > ```
> > >
> > > > **L2 (Intermediate):**
> > >
> > > > *Geo prompt:* "The overall appearance is a hemisphere with a cross-shaped recess. The main base feature is a hemisphere with a radius of 40mm. On the flat side, a deep cross-shaped groove is cut with a width of 10mm and depth of 20mm. The ends of the groove are open. The cross-shaped groove divides the hemisphere into four quadrant-like segments connected at the bottom."
> > >
> > > > *Seq prompt:* "Draw a semicircle with radius 40mm and rotate 180 degrees to generate a hemisphere. On the top plane, draw two rectangles that are concentric and perpendicular, both with width 10mm and lengths extending beyond the hemisphere edge. Extruded cut downward by 20mm to form a cross-shaped groove."
> > >
> > > > ```python
> > > > import cadquery as cq
> > > > sphere_radius = 40
> > > > slot_width = 10
> > > > slot_depth = 20
> > > > slot_length = 100
> > > > hemisphere = (
> > > >     cq.Workplane("XY")
> > > >     .sphere(sphere_radius)
> > > >     .cut(
> > > >         cq.Workplane("XY")
> > > >         .transformed(offset=(0, 0, -sphere_radius))
> > > >         .box(sphere_radius * 3, sphere_radius * 3, sphere_radius * 2, centered=True)
> > > >     )
> > > > )
> > > > ```
> > >
> > > > **L3 (Advanced):**
> > >
> > > > *Geo prompt:* "The overall shape is a slender spindle with cross-sections smoothly transitioning from circle to ellipse and back. The spine is controlled by a cubic Bezier curve with control points P0(0,0,0), P1(30,20,0), P2(100,25,0), P3(150,5,0). Four cross-sections: X=0 circle d=15mm, X=50 ellipse 25×20mm, X=100 ellipse 22×18mm, X=150 circle d=20mm. Shell wall thickness 1.5mm."
> > >
> > > > *Seq prompt:* "Create four datum planes at X=0,50,100,150. Draw respective circle/ellipse profiles on each. Draw a Bezier guide line with control points P0-P3. Loft through all sections with the guide line. Shell with 1.5mm wall thickness, removing both end faces."
> > >
> > > > ```python
> > > > import cadquery as cq
> > > > from math import isclose
> > > >
> > > > def solve_cubic_bezier_z_at_x(x_target, p0, p1, p2, p3, tolerance=1e-4):
> > > >     t_min, t_max = 0.0, 1.0
> > > >     for _ in range(100):
> > > >         t = (t_min + t_max) / 2
> > > >            ......
> > > >            ......
> > > > arm_solid = cq.Workplane("YZ").add(wires).toPending().loft(ruled=False)
> > > > result = arm_solid.faces("<X or >X").shell(-1.5)
> > > > ```
> > >
> > > > The progression is clear: L1 requires single primitives with basic features ; L2 introduces boolean operations and feature interactions ; L3 demands parametric curves, multi-profile loft, and shell operations. A qualitative leap in both geometric reasoning and code complexity.

---

### Official Review · Reviewer_12i8 · 2026-03-11

**Soundness:** 3
**Presentation:** 3
**Significance:** 2
**Originality:** 2
**Overall Recommendation:** 4
**Confidence:** 4

**Summary:**

This paper presents Text2CAD-Bench: a benchmark for evaluating text-toCAD models. It consists of four levels of difficulty. L1-L1 tests geometry fundamentals, L3 introduces complex operations that can alter topology, and L4 extends L3 to include real world models across a variety of domains. In total, the benchmark contains 600 models designed by AI with humans in the loop. The metrics are Chamfer Distance, Invalidity Rate, and IOU. They evaluate various LLMs, both general ones domain-specific models. They also ablate the choice of CAD representation, finding CadQuery to be a better choice to DeepCAD.

**Compliance With Llm Reviewing Policy:**

Affirmed.

**Final Justification:**

The authors addressed my concerns in the rebuttal so have raised my score to 4.

**Key Questions For Authors:**

I mentioned some questions in my weaknesses section.

**Limitations:**

Yes

**Strengths And Weaknesses:**

**Strengths**

This is an important topic area. CAD is something that is used everywhere in real-world design and AI has the potential to transform this practice. Having a good benchmark is essential and this work is a contribution in this domain. The benchmark is well motivated, the paper is easy to follow, and the level system is very useful. The results seem solid.

**Weaknesses**

 A major weakness to me is the validity of the metrics. Since we are only given text descriptions, there can be ambiguity over the desired result. A model may adhere to the prompt, but isn’t it possible that its output is sufficiently different so as to receive a low CD or IOU score? Are CD and IOU invariant to rigid transformations of the object? When I look at Fig 3, I see a lot of variations per-prompt and I am not sure whether these are invalid or not. The authors should discuss this. I also think some kind of semantic metric would be good, perhaps using a multimodal LLM as a judge, and showing that this metric aligns with human judgment.

Another concern I have is whether this work is suitable for ICML. The work seems valid, but there are not any modeling contributions and it seems better suited for a vision or graphics conference. That is just my opinion, and I am willing to change it if other reviewers find it suitable.

I also think there are some weaknesses with the presentation. Why is L4 not shown in Figure 2? Table 3 is also poorly designed. It does not show the two methods compared side-by-side.

---

> ### Author Rebuttal · Authors · 2026-03-31
>
> We thank Reviewer 12i8 for the constructive feedback and recognition of our work's importance. We address each concern below.
> ## Q1: Metric validity. Are CD and IoU invariant to rigid transformations? Can ambiguous descriptions cause valid outputs to receive low scores?
>
> We appreciate this important concern. Our evaluation pipeline addresses rigid transformation invariance through two mechanisms:
> (1) all models are normalized to unit bounding boxes before computing CD and IoU (Section 3.3.3), which eliminates translation and scale differences;
> (2) remaining rotational/mirror ambiguities are minimized by our design principle of "Unambiguous Ground Truth" in Section 3.1 to ensure  every description fully specifies all geometric parameters , leaving minimal room for valid alternative interpretations.
> Regarding the variations visible in Figure 3, the outputs that appear visually different from ground truth are indeed geometrically incorrect: they reflect models misinterpreting geometric specifications (e.g., wrong dimensions, missing features) rather than valid alternative solutions. We acknowledge that complete elimination of multi-solution ambiguity is difficult to guarantee, but we believe the impact is minimal given our rigorous specification and normalization procedures.
>
> ## Q2: A semantic metric using multimodal LLM-as-judge would be valuable.
>
> We agree that semantic evaluation provides a complementary perspective. In fact, for L4 (real-world applications), we adopted an LLM-based evaluation approach: we use GLM4.6V to score generated models based on 8 multi-view rendered images, evaluating both geometric fidelity (Geo Score) and task-specific feature completeness (Task Score) on a 0-10 scale across 5 structured questions per example. We will clarify this evaluation protocol in our revision with full details on the scoring rubric. Additionally, we  conduct human verification on a subset of L4 results to validate alignment between LLM-based and human judgments, and will review L3 results to ensure evaluation reliability.
>
> ## Q3: Whether this work is suitable for ICML given no modeling contributions.
>
> We respectfully argue that our work fits well within ICML's scope. As the first dedicated benchmark for text-to-CAD generation, our work establishes the evaluation infrastructure that this emerging field currently lacks. First, Text2CAD-Bench evaluates the end-to-end capability from text to CAD models, which involves a comprehensive ability chain of language understanding, geometric reasoning, and code generation — core concerns of the ML community. Second, ICML has a strong tradition of accepting high-quality benchmark papers (e.g., HumanEval, GAIA) that diagnose capability boundaries and drive research progress without necessarily proposing new models.
> Moreover, our benchmark reveals several non-obvious findings of broad interest:
> (1) geometric descriptions outperform command sequences on L1-L2, contradicting the prevailing paradigm where prior works (Text2CAD, CAD-Coder, Text2CADQuery) exclusively adopt sequence-based descriptions;
> (2) ranking reversals on complex tasks (e.g., Claude surpassing GPT-5.2 on L3) cannot be trivially extrapolated from simple-task performance;
>  (3) domain-specific models exhibit a striking IR-CD contradiction (lowest IR yet highest CD);
> (4) on L4, MiniMax achieves the highest design quality scores despite having the worst IR for 83%, while GPT-5.2 — the strongest model on L1-L3 geometric precision — scores lowest in Table 2. This reveals that code executability, geometric precision, and design intent understanding are three independent capability dimensions, reinforcing the necessity of multi-dimensional evaluation.
>
> ## Q4: L4 is not shown in Figure 2 & Table 3 does not show methods side-by-side.
>
> Figure 2 visualizes Chamfer Distance across benchmark levels for quantitative comparison. Since L4 employs a qualitative evaluation framework (Geo Score and Task Score via GLM4.6V assessment) that differs fundamentally from the L1-L3 quantitative metrics, combining them in one figure would be misleading. L4 results will presented separately in Table 2. We will improve the presentation to make this distinction clearer and we will redesign Table 3 to display  results side-by-side for direct comparison.

---

> > ### Author Rebuttal · Reviewer_12i8 · 2026-04-02
> >
> > The authors have provided answers that are mostly satisfactory to my concerns. I will raise my score to 4.

---

### Official Review · Reviewer_vFhm · 2026-03-11

**Soundness:** 2
**Presentation:** 3
**Significance:** 2
**Originality:** 2
**Overall Recommendation:** 4
**Confidence:** 4

**Summary:**

This paper proposes Text2CAD-Bench, a benchmark for evaluating text-to-CAD generation methods. Specifically, the benchmark has 600 human-curated examples across four complexity levels, with increase geometry complexity. Each example is paired with dual-style prompts, a geometric description for non-expert users and sequence descriptions mimicking expert CAD conventions. The authors evaluate seven general-purpose LLMs and three domain-specific models with metrics like Chamfer Distance, Invalidity Rate, and IoU.

**Compliance With Llm Reviewing Policy:**

Affirmed.

**Final Justification:**

Most of my concerns have been addressed, so I've raised my score.

**Key Questions For Authors:**

- How are the Geo Score and Task Completion Score in Table 2 computed? If these are LLM-evaluated scores, which model serves as the judge, and how was the rubric validated?

- For the L3 invalidity rates near 70–90%, does the retry prompt contribute significantly to getting these numbers below 100%, or do most models fail even after the retry?

**Limitations:**

Yes

**Strengths And Weaknesses:**

**Strengths**:
- The paper is well-written and easy to follow. The four-level complexity hierarchy and dual-prompt design are clearly motivated.
- The idea of stratifying benchmark difficulty by geometric operation complexity is intuitive and addresses a real gap in prior work.

**Weaknesses**:
- The L4 evaluation uses a different metric (Geo Score and Task Completion Score) compared to L1-L3 (CD, IR, IoU), but the paper does not explain how these scores are computed. It will be beneficial to describe this clearly.
- L4 ranking in Table 2 is inconsistent with L1-L3 performance. GPT-5.2 achieves the lowest CD in Table 1 but receives the lowest Geo/Task scores in Table 2, while MiniMax and Qwen score much higher. Without a clear explanation of the L4 metric, this inconsistency undermines trust in the results.
- CAD-Coder is mentioned in Section 2.2, but it is not included in the evaluation. It will be helpful to compare with CAD-Coder to better position the benchmark.
- The ablation study in Table 3 only covers two models (Gemini-3-Flash and Qwen3-max) for the command sequence comparison. Including more models (e.g., GPT-5.2 or Claude-4.5) would strengthen the conclusion that CadQuery is superior.
- The contribution of this paper is primarily a curated dataset paired with an evaluation of existing models. The main findings, such as most models degrade on complex geometry and that CadQuery outperforms command sequences, are largely expected and not surprising. For a top conference like ICML, it is unclear whether a benchmark paper of this scope (600 examples, single CAD framework, no new model or training methodology) meets the bar without a stronger set of non-obvious insights.

---

> ### Author Rebuttal · Authors · 2026-03-31
>
> We thank Reviewer vFhm for the detailed and constructive feedback. We address each concern below.
>
> ## Q1: How are the Geo Score and Task Completion Score in Table 2 computed? If LLM-evaluated, which model serves as judge and how was the rubric validated?
>
> We appreciate this question and apologize for the insufficient explanation. L4 evaluation uses GLM4.6V as the judge model. For each generated CAD model, we render 8 multi-view images and present them alongside 5 structured questions covering: overall structural correctness, major feature presence, dimensional accuracy, detail completeness, and functional suitability (each scored 0-10). The Geo Score reflects geometric appearance; the Task Score captures feature completeness. We also report L4 Invalidity Rates, revealing striking patterns: MiniMax has the highest IR (83%) yet the highest Geo/Task scores, while Gemini has the lowest IR (25%), demonstrating that executability and design quality are distinct dimensions. We will include the full rubric in our revision and conduct human verification on a subset.
>
> ## Q2: L4 ranking is inconsistent with L1-L3 performance. GPT-5.2 achieves the lowest CD in Table 1 but receives the lowest scores in Table 2.
>
> Our L4 IR data further illuminates this inconsistency.
> Three independent capability dimensions emerge: Code executability — Gemini achieves the lowest L4 IR (25%); Geometric precision — GPT-5.2 leads on L1-L3 CD; Design intent understanding — MiniMax scores highest on Geo/Task despite the worst IR (83%), meaning its few successful outputs capture application features exceptionally well. GPT-5.2, despite superior geometric precision, scores lowest in Table 2 — it fails to interpret domain conventions in real-world contexts. This three-way divergence demonstrates that no single metric captures text-to-CAD capability, reinforcing our multi-dimensional evaluation design.
>
> ## Q3: CAD-Coder is mentioned in Section 2.2 but not included in evaluation.
>
> We investigated including CAD-Coder. However, the text-to-CAD version of CAD-Coder has not been open-sourced; the publicly available repository under the same name is a different work focused on image-to-CAD generation. We were therefore unable to reproduce their results on our benchmark. We will clarify this in our revision.
>
> ## Q4: Table 3 ablation only covers two models. Including more models would strengthen the conclusion.
>
> We agree, and have now completed additional ablation experiments with GPT-5.2 and Claude-4.5-Sonnet. The results reinforce our original conclusion with greater nuance: on L1, geometric prompts achieve significantly lower IR (~15%) across all four models, while CD and IoU remain comparable — indicating that geometric descriptions primarily improve code executability at the basic level. On L2, geometric prompts maintain the IR advantage while also yielding notably better CD and IoU, demonstrating that as geometric complexity increases, the benefit of geometric descriptions extends from executability to geometric fidelity. These expanded results will be presented in the revised Table 3.
>
> ## Q5: Contribution scope — whether a benchmark of 600 examples with no new model meets the ICML bar.
>
> We respectfully argue that our benchmark provides contributions beyond dataset curation, revealing several non-obvious findings:
> First, geometric descriptions outperform command sequences on L1-L2. This contradicts the prevailing paradigm: prior works (Text2CAD, CAD-Coder, Text2CADQuery) exclusively adopt sequence-based descriptions, and even our team initially expected sequences to be superior.
> Second, ranking reversals on complex tasks — Claude surpassing GPT-5.2 on L3 despite trailing on simpler tasks — show that simple-geometry capabilities do not extrapolate.
> Third, and most strikingly, our L4 results reveal that code executability, geometric precision, and design intent understanding are three independent capability dimensions. MiniMax achieves the highest design quality despite the worst IR (83%), while GPT-5.2 — the L1-L3 precision leader — scores lowest on L4. Domain-specific models show the mirror pattern: lowest IR yet highest CD. No prior work has demonstrated this three-way decoupling. These insights emerged precisely because of our systematic design (four complexity levels × dual prompt styles × multi-metric evaluation).
>
> ## Q6: For L3 invalidity rates near 70-90%, does the retry prompt contribute significantly?
>
> Our evaluation employs up to 3 retry attempts with error feedback. The retry mechanism provides modest improvements — typically reducing IR by 5-15 percentage points — but most models still fail on L3 even after retries, confirming that the high invalidity rates reflect fundamental capability limitations rather than recoverable errors. We will include detailed per-retry statistics in our revision.

---

> > ### Author Rebuttal · Reviewer_vFhm · 2026-04-05
> >
> > Most of my concerns are solved.

---

> > > ### Author Response · Authors · 2026-04-08
> > >
> > > > We sincerely appreciate  reviewer vFhm's positive acknowledgement. As a brief update, we have completed the expanded ablation (Q4) with GPT-5.2, Claude-4.5-Sonnet, DeepSeek-V3.2, MiniMax M2.5, and GLM4.7 — covering all seven general-purpose LLMs. The results consistently confirm our conclusion: on L1, geometric prompts reduce IR  across all models while CD and IoU remain comparable; on L2, geometric prompts additionally show clear CD and IoU advantages. The expanded results will appear in the revised Table 3.

---

### Official Review · Reviewer_WnWU · 2026-03-13

**Soundness:** 2
**Presentation:** 3
**Significance:** 3
**Originality:** 3
**Overall Recommendation:** 3
**Confidence:** 4

**Summary:**

The paper introduces Text2CAD-Bench, a new benchmark for generating parametric CAD models from text. It contains 600 human-curated examples across four difficulty levels, ranging from basic geometry to complex topology and real-world domains. Each example includes both geometric prompts and procedural sequence prompts to test different styles of text input. The experiments show that current LLMs and domain-specific models work reasonably on simple cases but struggle on harder CAD tasks. Overall, the benchmark is positioned as a more realistic and challenging testbed for future text-to-CAD research.

**Compliance With Llm Reviewing Policy:**

Affirmed.

**Final Justification:**

I will maintain my scores as I find claims made in rebuttal lacking evidence.

**Key Questions For Authors:**

1. While the benchmark setup is interesting, I have a central doubt that the evaluation is highly sensitive to textual formulation. The authors’ explanation that sequence prompts better capture complex CAD procedures at L3 is plausible, but it is argued mainly through CD, which measures local surface similarity. Since the benchmark also uses IR and IoU to assess executability and global shape agreement, this analysis/comparison seems too narrow to establish that the effect reflects underlying text-to-CAD ability rather than prompt/template sensitivity. For a benchmark paper, the key question is whether results are stable across reasonable ways of expressing the same task; the Geo-vs-Seq comparison suggests they may not be. Without stronger ablations on description style, it remains unclear whether the benchmark cleanly measures CAD understanding or partly rewards alignment to a particular prompt format.

2. A related concern is the interpretation of IR for Text2CAD. The paper notes that sequence descriptions extend the expert-level instruction format introduced in Text2CAD. My concern is - Text2CAD’s substantially lower IR may not only reflect superior robustness in CAD code generation, but also a distributional advantage because the benchmark’s sequence descriptions are derived from a format closely related to its training representation. To address this issue, the authors should consider explicitly discussing the potential alignment between their sequence-description format and Text2CAD’s training representation and add controlled ablations with alternative sequence templates or paraphrases.

3. In the experiment design: the number of attempts for The use of official APIs without hyperparameter tuning is understandable from an off-the-shelf evaluation perspective. However, given that the benchmark is already sensitive to prompt style, it would strengthen the paper to show that the main conclusions are also robust to reasonable inference settings for strong LLM baselines (e.g., temperature, formatting constraints, or simple repair/retry policies). Otherwise, it is difficult to know whether some of the reported differences reflect underlying model capability or incidental choices in evaluation protocol.

4. Table 2 is under-explained. It is not clearly discussed in the main text, and the Geo/Task scores are not sufficiently specified in terms of computation or evaluation protocol. This makes the table difficult to interpret and weakens its contribution to the empirical case made by the paper.

5. The 600 human-curated questions are potentially useful, but the paper does not clarify how distinct they are from existing datasets or common prompt templates. While the reported results do not suggest an obviously saturated benchmark, some analysis of overlap or contamination risk would still be important for establishing benchmark validity.

**Limitations:**

yes

**Strengths And Weaknesses:**

Strength:
1. A standardized evaluation suite for text-to-CAD is potentially valuable to the community.
2. The benchmark includes multiple difficulty levels and prompt styles
3. The benchmark is rich in fine details such as context and diverse examples
4. Human expert curated dataset which is a critical and positive aspect

Weakness:
1. Only 100 examples for real-world application
2. The benchmark appears sensitive to prompt style and description ordering
3. Grammatical and punctuation errors in the paper
4. The dataset curation process is better described than quantified through means of any statistics

---

> ### Author Rebuttal · Authors · 2026-03-31
>
> We thank Reviewer WnWU for the insightful and rigorous review. We address each concern below.
> ## Q1: Evaluation sensitivity to textual formulation. The Geo.vs Seq. comparison is analyzed mainly through CD, which is too narrow. Without stronger ablations on description style, it is unclear whether the benchmark measures CAD understanding or prompt format sensitivity.
>
> We appreciate this concern. We acknowledge that prompt style does influence results, and that this effect intensifies with geometric complexity — L1-L2 show relatively stable model rankings across prompt styles, while L3 exhibits notable volatility. However, we argue this is not a benchmark deficiency but rather a meaningful phenomenon that our dual-style design is specifically intended to expose.
> In real-world usage, users with different expertise naturally describe the same geometry differently. That models respond differently — and that the advantage shifts from geometric descriptions to sequence descriptions is a practically meaningful finding.
> We agree that our current analysis relies too heavily on CD. We will supplement with IR and IoU cross-analysis across prompt styles. From Table 1, the IR pattern corroborates the CD findings: on L1-L2, geometric prompts generally yield lower IR, while on L3, the difference narrows or reverses for several models. We will present this multi-metric analysis in our revision.
>
> ## Q2: Text2CAD's lower IR may reflect distributional advantage rather than robustness, since sequence descriptions are derived from a format close to its training representation.
>
> This is a valid concern. However, we note that Text2CAD exhibits the same pattern under geometric prompts — which differ substantially from its training format. Specifically, Text2CAD's IR under geometric prompts (11.0% on L1, 6.0% on L2, 2.0% on L3) is comparable to its IR under sequence prompts (8.0%, 5.0%, 6.0%). If low IR were purely due to distributional alignment with sequence-style inputs, we would expect a significant IR increase under geometric prompts, which we do not observe.
> Simultaneously, Text2CAD's CD remains extremely high (>220) regardless of prompt style, confirming that low IR does not indicate geometric correctness. We discuss this IR-CD contradiction in Section 4.2.1, highlighting it as evidence for the limitations of single-metric evaluation. We will add an explicit discussion of the distributional alignment concern as suggested.
>
> ## Q3: Robustness to inference settings . It is difficult to know whether reported differences reflect model capability or evaluation protocol choices.
>
> Our evaluation employs up to 3 retry attempts with error feedback, which we will state more explicitly. Using official API default settings follows the off-the-shelf evaluation paradigm of established benchmarks (HumanEval, MATH-500), ensuring reproducibility and reflecting practical deployment conditions.
>
> ## Q4: Table 2 is under-explained. Geo/Task scores are not sufficiently specified.
>
> We apologize for this omission. L4 uses GLM4.6V as judge, scoring via 8 multi-view images across 5 structured questions. Geo Score evaluates geometric appearance; Task Score evaluates feature completeness. We will provide the full rubric and validate with human judgments. Our newly compiled L4 IR data further enriches the picture: MiniMax has the highest IR (83%) yet the highest Geo/Task scores, while Gemini has the lowest IR (25%). Combined with GPT-5.2 leading L1-L3 precision but scoring lowest on L4. As detailed in our response to Reviewer vFhm, we believe code executability, geometric precision, and design intent understanding)decoupling is itself a significant finding.
>
> ## Q5: The 600 examples — how distinct are they from existing datasets? Overlap or contamination risk should be analyzed.
>
> We address this from multiple angles. First, our examples require advanced operations  that are entirely absent from existing datasets — DeepCAD and Text2CAD are limited to sketch-extrude sequences. This fundamental difference in geometric coverage makes contamination at the task level hard for our most challenging examples. Second, our CadQuery ground-truth code is in a completely different language from DeepCAD's DSL, eliminating code-level overlap. Third, our descriptions are human-authored following specific templates (Section 3.2), whereas Text2CAD uses automated generation from command  sequences — the authoring processes are fundamentally different. All examples undergo independent human verification for quality and uniqueness. We have conducted 5-gram overlap analysis between our prompt texts and text2cad dataset descriptions. The results confirm minimal textual overlap: mean Jaccard-5 scores are below 0.006 and mean Containment-5 scores are below 0.022 for both L1 and L2, with even per-example maximums remaining low (all below 0.11). This provides strong quantitative evidence against contamination concerns.

---

> > ### Author Rebuttal · Reviewer_WnWU · 2026-04-02
> >
> > I thank the authors for proving clarifications to my concern. I find most of them satisfactory.
> > The rebuttal to Q1 lacks the additional IR and IoU cross-analysis across prompt styles. It's still unclear as to what extent the benchmark reflects prompt sensitivity or CAD understanding. An updated Table 1 and 2 would have helped support the claims.
> > I will maintain my current scores.

---

> > > ### Author Response · Authors · 2026-04-08
> > >
> > > We thank reviewer WnWU for the follow-up. We provide the IR and IoU cross-analysis across prompt styles from Table 1. The IR pattern strongly corroborates the CD findings: on L1-L2, geometric prompts yield lower IR for nearly all general-purpose LLMs (typically 5-10 %), with the gap narrowing on L3. For IoU, performance is comparable across both prompt styles on L1, while geometric prompts begin to show advantages on L2 for most models. Notably, IoU on L1 is comparable across both prompt styles, while IR already shows a clear Geo advantage — which validates our multi-metric design: different metrics capture different dimensions of the Geo-vs-Seq distinction. Moreover, our core findings — such as ranking reversals between L2 and L3, and the three-way decoupling on L4 — involve large performance gaps that are unlikely to be artifacts of prompt format sensitivity. We will incorporate this full cross-analysis into the revised manuscript.

---

### Decision · Program_Chairs · 2026-04-30

**Decision:**

Reject

**Comment:**

The paper ended up with borderline scores (WAs and WRs). The rebuttal phase involved discussion among the reviewers and authors. Concerns remained about whether the paper is of interest to an ML conference; there were concerns about the benchmark and the lack of sufficient evaluation -- two of the reviewers remained on the negative side. Given that the topic is not new (no clear new problem or solution is being proposed) and no one championed the work, the AC recommends rejecting the current submission.